# Microplastic burden in marine benthic invertebrates depends on species traits and feeding ecology within biogeographical provinces

Adam Porter [1] ✉, Jasmin A. Godbold [2], Ceri N. Lewis [1], Georgie Savage [1], Martin Solan [2] & Tamara S. Galloway[1]

The microplastic body burden of marine animals is often assumed to reflect levels of environmental contamination, yet variations in feeding ecology and regional trait expression could also affect a species' risk of contaminant uptake. Here, we explore the global inventory of individual microplastic body burden for invertebrate species inhabiting marine sediments across 16 biogeographic provinces. We show that individual microplastic body burden in benthic invertebrates cannot be fully explained by absolute levels of microplastic contamination in the environment, because interspecific differences in behaviour and feeding ecology strongly determine microplastic uptake. Our analyses also indicate a degree of species-specific particle selectivity; likely associated with feeding biology. Highest microplastic burden occurs in the Yellow and Mediterranean Seas and, contrary to expectation, amongst omnivores, predators, and deposit feeders rather than suspension feeding species. Our findings highlight the inadequacy of microplastic uptake risk assessments based on inventories of environmental contamination alone, and the need to understand how species behaviour and trait expression covary with microplastic contamination.

Despite ambitious waste management strategies designed to reduce plastic litter[1], it is anticipated that the introduction of plastic to the marine environment will continue to rise for decades[2], exacerbating any effects that plastic pollution may have on species and ecosystems[3,4]. Like all particulate matter, the fate of marine microplastic (<5 mm) is to sink to, and accumulate on the seafloor[5,6], a habitat that harbours high levels of biodiversity[7,8]. As these accumulations of microplastics comprise a complex mix of heterogeneous particles with a range of shapes, sizes, colours, polymers, and additives[9,10] that match the size spectrum of typical prey items and food parcels[11,12], they are bioaccessible to a range of benthic invertebrates[13,14]. Mean microplastic concentrations in soft sediment habitats can be close to (continental slope, 502 microplastic kg⁻¹), or greatly exceed (hadal trenches, 2782 microplastics kg⁻¹ [15]) estimated safe limits (540 microplastic particles kg⁻¹ [16]), particularly in areas that accumulate organic matter[17]. However, the uptake (here defined as adhesion, entanglement, and/or ingestion) of microplastic by species is unlikely to be a universal function of absolute levels of microplastic contamination or inter-specific differences in body size[18,19] because the way in which species interact with the sediment environment is highly dependent on taxonomic position[13,20,21], feeding and foraging strategy and individual species behaviours[13,20,22,23], all of which can be population dependent and modified by abiotic (nutrient enrichment[24]; flow[25]; temperature[26]) and/or biotic (e.g. predation[27]) circumstances[28]. Even

[1]Department of Biosciences, University of Exeter, Geoffrey Pope Building, Exeter EX4 4QD, UK. [2]School of Ocean and Earth Science, National Oceanography Centre Southampton, University of Southampton, Waterfront Campus, Southampton SO14 3ZH, UK. ✉e-mail: a.porter@exeter.ac.uk

closely related species can behave differently[29], such that realised levels of microplastic uptake can be highly variable both within and between species[30–32].

Despite growing knowledge of the ecological effects of microplastic[3,33], assessment of the risks posed by exposure and uptake is hindered by major gaps in our understanding of when, how and which species are most likely to interact with microplastic[10,34]. Resolving species-microplastic interactions is fundamental to setting appropriate contamination thresholds[35], designing innovative solutions and predicting the most likely ecological consequences of microplastic contamination, but the relationship between the functional role of species and microplastic contamination at regional to global scales remains unknown. Here, we combine a comprehensive global inventory (55 studies; 244 locations, 412 unique observations; 69.08˚N–73.49˚S, 171.15˚W–170.22˚E) of microplastic body burden in sediment-dwelling marine invertebrates, with taxonomic information, to investigate whether global patterns of microplastic burden are associated with species traits (size, habit, mobility, feeding type, and environmental positioning) and/or differ across 16 biogeographic provinces[36,37]. Our focus was to establish whether commonly used taxonomic considerations of risk provide consistent and relevant

information that will reduce uncertainty in projecting which species are most at risk of microplastic exposure[38,39]. Given the diversity of feeding modes, we expected that gut morphologies and gut retention times in marine invertebrates[40,41], as well as changes in species behaviour that depend on the biotic and/or environmental conditions they experience[28,42,43] would be important in determining microplastic burden. We demonstrate that feeding mode (in particular predatory, omnivorous, and deposit-feeding organisms), rather than environmental microplastic loading, coupled with geographical location, determine invertebrate microplastic body burden.

## Results

### Global inventory of invertebrate microplastic burden

Our analysis reveals that records of microplastic burden are distributed across 16 biogeographical provinces, but that there are substantive gaps in spatial and seasonal coverage of invertebrate microplastic burden across all major oceans, and a global paucity of data beyond shelf depth (Fig. 1). The data showed a strong bias towards the Northern Hemisphere and, in particular, the coastal regions of North America, Europe and Southeast Asia. Remote locations, such as the Pacific, South Atlantic, Indian Ocean and the Poles, were either

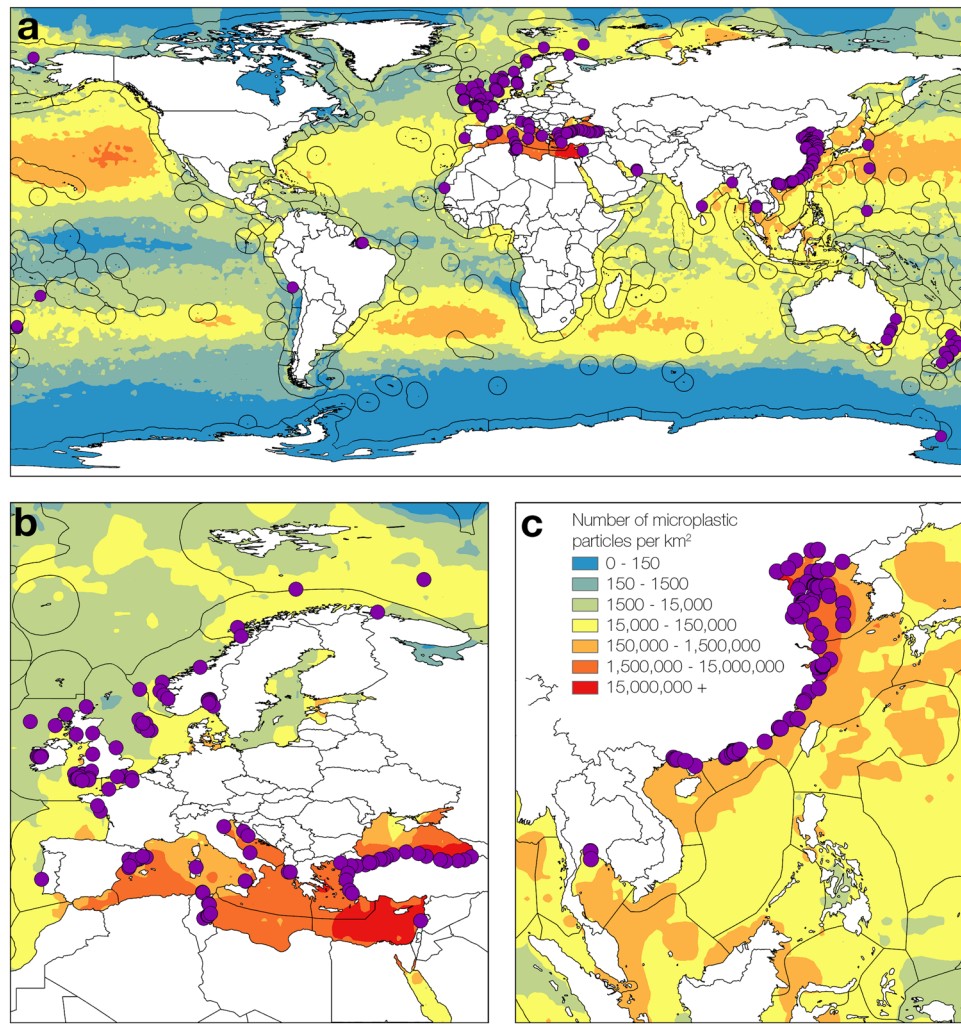

**Fig. 1 | A map to show the global spread of marine benthic invertebrates burdened with microplastics.** (**a**) The global distribution includes 412 study locations (purple dots) that report the presence of microplastic in marine benthic invertebrates. Detailed panels of the North-East Atlantic and Mediterranean (**b**) and (**c**) South-East Asian regions show data spread. Colour shading depicts model[100] predictions of microplastic particle distribution (number per km²; key located in China (Panel **c**)) and boundaries of biogeographical provinces ([36,37] ocean boundary lines) are indicated. Maps were drawn using ArcGIS Desktop[105] and country boundaries are provided by ESRI using data from Esri; Garmin International, Inc.; U.S. Central Intelligence Agency (The World Factbook); National Geographic Society.

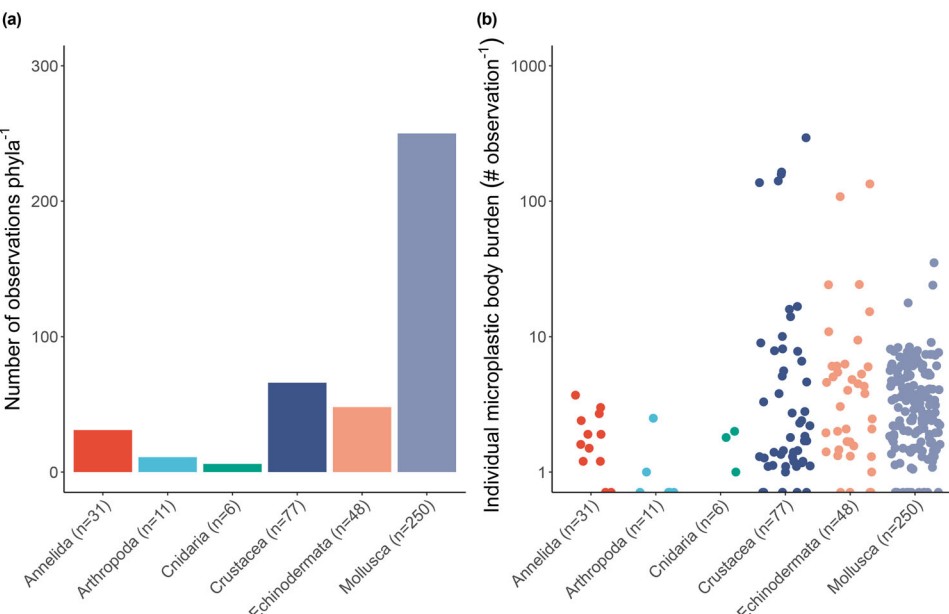

**Fig. 2 | A summary of the marine benthic invertebrates investigated for microplastic burden globally.** The (**a**) number of individual records (observations) and (**b**) reported values of individual microplastic body burden (mean number of microplastic particles ind.$^{-1}$, horizontally jittered for clarity) are presented for each Phylum.

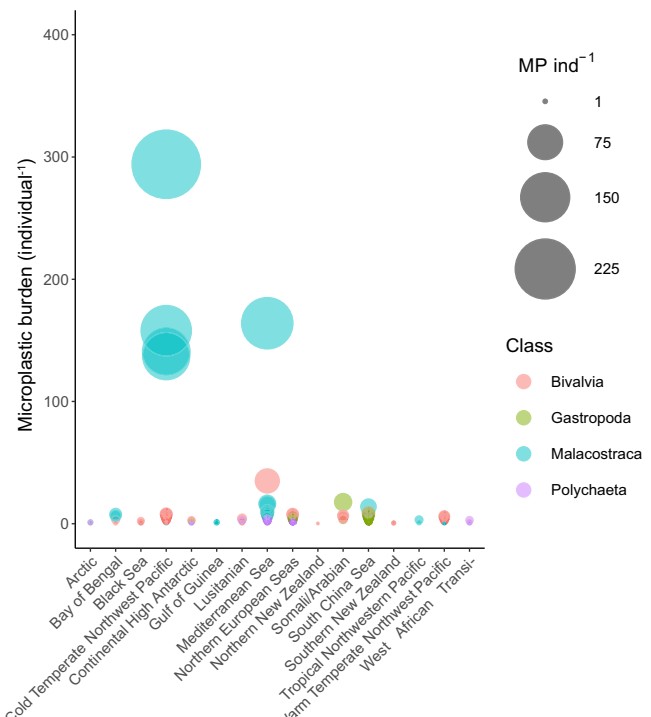

**Fig. 3 | The global distribution of individual microplastic body burden for the four main taxonomic classes.** The individual microplastic body burden individual $^{-1}$ (MP ind$^{-1}$) (bubble diameter) collected from the available literature for each Class (colour) with ≥ 30 observations in each of the 16 biogeographic provinces where data were available.

inadequately constrained or lack information. Nevertheless, taxonomic coverage within the 8353 sampled individuals was reasonable, representing 141 species across 6 phyla (Mollusca, 57.4%, $n = 4799$; Crustacea, 16%, $n = 1338$; Echinodermata, 10.5%, $n = 880$; Cnidaria, 4%, $n = 338$ Annelida, 9.4%, $n = 786$; Arthropoda, 2.5%, $n = 212$; Fig. 2).

The presence of microplastic was common (microplastic burden >0 in 93% observations) across all phyla, but highly variable (mean particles ind.$^{-1}$ = 0, $n = 29$; mean particles ind.$^{-1}$ = <1, $n = 121$; mean particles ind.$^{-1}$ = 1–294, mean 8.3 ± 1.70; median, 2.95, $n = 262$) (coefficient of variation: Mollusca, 124.22%; Crustacea, 307.30%; Echinodermata, 277.12%; Cnidaria, 74.99%; Annelida, 90.85%; Arthropoda, 160.27%), indicating that the risk of uptake to microplastic differs between individuals and between taxonomic groups. Within quality control guidelines (see methods and Supplementary Table 1), we found ~2230 microplastic particles, of which 51.4% was measured within Crustacea, followed by the Echinodermata (30%), Mollusca (9.2%), Annelida (6.8%), Arthropoda (1.5%) and Cnidaria (1.2%). The highest burden of microplastic was found in decapod crustaceans, *Crangon affinis*, collected from the South Yellow Sea, China (294 microplastics ind.$^{-1}$), and *Aristeus antennatus*, from the NW Mediterranean Sea (164 microplastics ind.$^{-1}$). The latter exclusively contained fibrous microplastic particles.

### Patterns of invertebrate microplastic burden
To separate any effect of broad habitats and species lifestyles on microplastic body burden, whilst retaining sufficient discriminatory power, we used Spalding's biogeographical province and Class.

### Effects of biogeographical province
Invertebrate microplastic body burden differed with the biogeographic province (L-ratio = 138.359, d.f. 15, $p < 0.0001$; Fig. 3); individuals collected from the Cold Temperate Northwest Pacific (Yellow Sea [56 locations] and Japan Trench [1 location]) had the highest mean (±s.e.) individual microplastic body burden (25.41 ± 10.71 ind$^{-1}$), followed by the Mediterranean Sea (8.18 ± 4.02 ind$^{-1}$, 41 locations, incorporating the Eastern coastline of Spain across to the Turkish Aegean Sea), whilst individuals collected from Northern New Zealand (0.15 ± 0.15 ind$^{-1}$, 2 locations) and Southern New Zealand (0.24 ± 10.71 ind$^{-1}$, 14 locations) showed the lowest mean individual microplastic body burden.

### Effects of Class
There were consistent differences in individual microplastic body burden between the four classes in our study that included more than

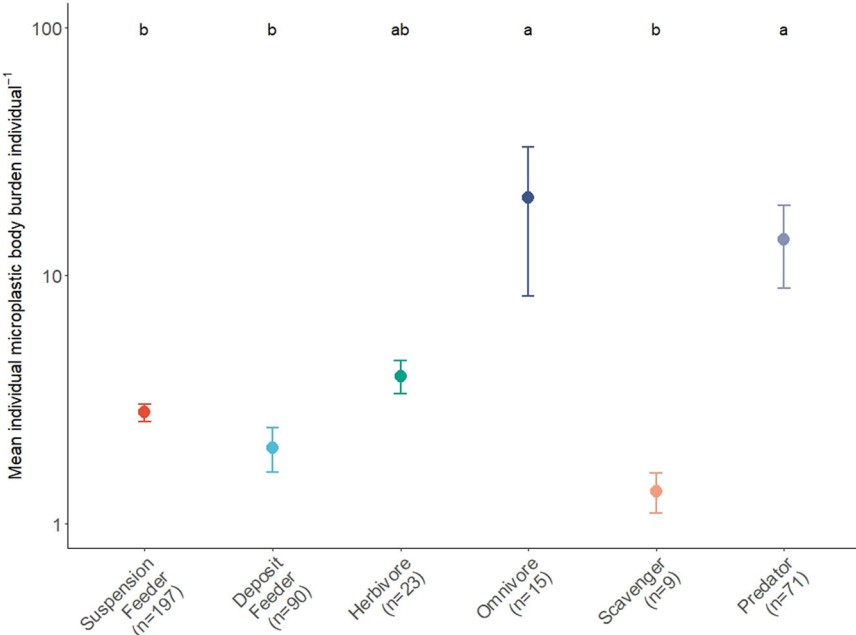

**Fig. 4 | Observed mean (± s.e.) microplastic burden individual⁻¹ for contrasting feeding modes of marine benthic invertebrates.** The statistical method used was a one-way ANOVA, with a Tukeys post-hoc test; (F$_{(6, 405)}$ = 4.182, $p$ < 0.001).

Statistically significant groupings are indicated by letters. Different colours are used only for illustrative purposes to indicate different feeding modes.

30 observations (L-ratio = 24.957, d.f. = 3, $p$ < 0.0001; Fig. 3). The highest mean (±s.e.) individual microplastic body burden occurred in the Malacostraca (15.44 ± 5.96 ind⁻¹), followed by the Bivalvia (2.88 ± 0.24 ind⁻¹), Gastropoda (2.23 ± 0.41 ind⁻¹), and Polychaeta (1.02 ± 0.17 ind⁻¹).

Furthermore, when analysing the full dataset, save for the Malacostraca, the faunal classes identified to have significantly higher body burdens of plastic were the Asteroidea, Cephalopoda, Echinoidea, Holothuroidea, and Ophiuroidea (Fig. S1) which were also those underrepresented in the literature (<14 observations per Class). Indeed, of the 29 non-symbiont phyla that exist in marine systems[44,45], only 5 (Mollusca, Echinodermata, Cnidaria, Crustacea, Annelida) are represented, constituting a significant knowledge gap.

### Effects of species traits

We investigated the role of species traits in determining whether the patterns we observe in invertebrate microplastic body burden were constrained by taxon-specific physiological or morphological limitations imposed by phylogenetic history or by the functional role of individuals, irrespective of taxonomic identity. We found that individual microplastic body burden was dependent on feeding mode (F$_{6,358}$ = 4.41, $p$ < 0.001), but not position within the sediment (F$_{2,358}$ = 1.19, $p$ = 0.31), mobility (F$_{5,358}$ = 0.69, $p$ = 0.60), habit (F$_{3,358}$ = 0.11, $p$ = 0.97), or the wet weight of an organism (F$_{1,358}$ = 0.59, $p$ = 0.45). The highest mean (±s.e.) individual microplastic body burden occurred in omnivores (20.72 ± 12.42 ind⁻¹, $n$ = 15) and predators (15.11 ± 5.57 ind⁻¹, $n$ = 71), followed by herbivores (4.16 ± 1.03 ind⁻¹, $n$ = 11), grazers (3.72 ± 0.7 ind⁻¹, $n$ = 12), suspension feeders (2.80 ± 0.23 ind⁻¹, $n$ = 197), deposit feeders (2.03 ± 0.41 ind⁻¹, $n$ = 90), and scavengers (1.35 ± 0.31 ind⁻¹, $n$ = 15) (Fig. 4). However, mean individual microplastic body burden was associated with some variance, often driven by species with disproportionately high microplastic body burdens, in particular, the predators (CV = 310%), omnivores (CV = 232%) and deposit feeders (CV = 194%), relative to the remaining groups (suspension feeders, 117%; scavengers, 89%; herbivores, 82% and grazers, 65.3%). Indeed, the highest observed individual microplastic body burden counts represented a limited number of species

(omnivores, the decapod, *Oregonia gracilis*, from the Yellow Sea, 141 and 137 particles ind.⁻¹; predators, the Crustaceans, *Crangon affinis*, 294 particles ind.⁻¹ and *Romaleon gibbosulum* 158 particles ind.⁻¹, the Echinoderms, *Ophiura sarsii*, 108 particles ind.⁻¹ and *Luidia quinarian*, 134 particles ind.⁻¹ and, from the Mediterranean off the coast of Barcelona, the Crustacean *Aristeus antennatus*, 164 particles ind.⁻¹). Removal of these high counts from the analyses reduced mean individual microplastic body burden for the predators (3.26 ± 0.6 s.e. particles ind.⁻¹) and omnivores (2.52 ± 0.76 s.e. particles ind.⁻¹). In contrast, although there were only 15 records and 3 species (the amphipods, *Hirondellea dubia*, *Hirondellea gigas*, and *Eurythenes gryllus*), scavengers were found to have the least individual microplastic body burden (range of 0.9–3.3 particles ind.⁻¹). Deposit feeders show high variance, despite a comparatively low mean (±s.e.) individual microplastic body burden (2.03 ± 0.41 ind.⁻¹). For example, there were 6 species (2 molluscs and 4 crustacea) with no microplastic body burden, but also 5 observations of the Japanese sea cucumber (*Apostichopus japonicus*) with an individual microplastic body burden ranging from 6.08–24.2 microplastic particles ind.⁻¹ (Fig. 4). Overall, there were consistent effects of feeding traits on individual microplastic body burden, but with varying degrees of confidence across taxonomic and functional groupings that most likely relate to differences in sampling effort.

### Relationship between risk factors

Geographic location, Class, and feeding mode were correlated with, and are therefore, potential drivers for, individual microplastic body burden. Geographical location explained the most variance (11%), followed by Class and feeding mode, each explaining ~6% of the data. In addition, there was a significant interaction between feeding mode and province, feeding mode and Class, and province and Class.

### Effects of microplastic shape, size and colour

Most observations ($n$ = 412) reported the shape (93%), size (90%), colour (57%) and polymer type (88%) of the recovered microplastic. Our results showed that fibres were the most commonly reported shape (99% of studies), followed by fragments (66%), pellets (32%),

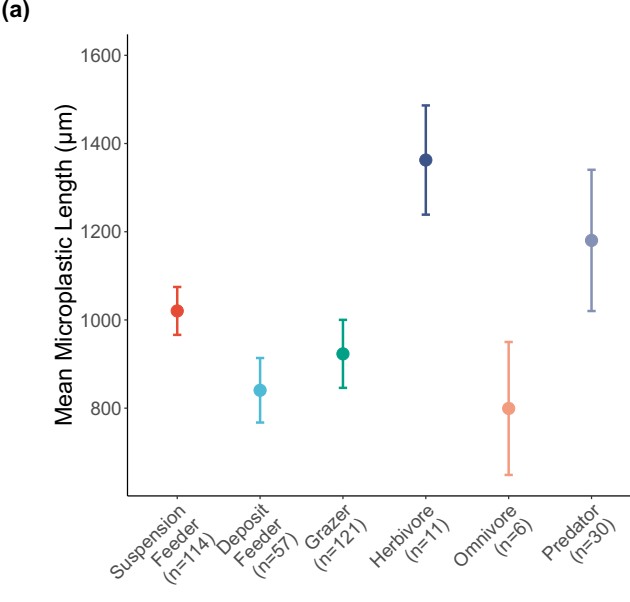

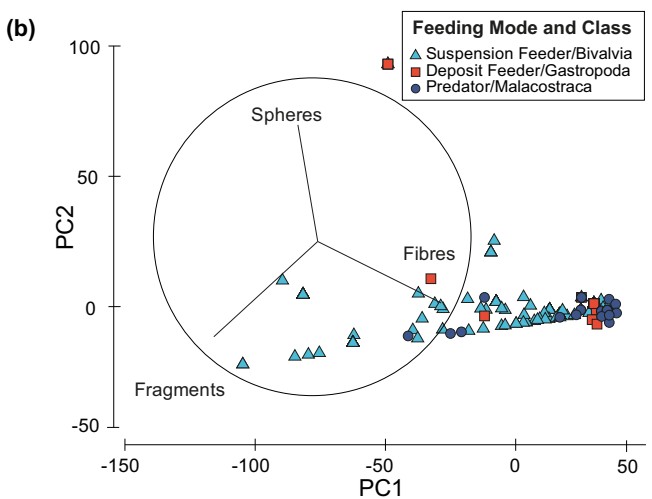

**Fig. 5 | A summary of microplastic characteristics within invertebrate feeding mode and Class for all groups with ≥30 observations.** (a) The mean (± s.e.) microplastic length for each feeding mode and (b) Principal Component Analysis microplastic shape demonstrating the affinity of feeding modes/Class to shape in marine benthic invertebrates. Overall, there were more fibres in all feeding modes, especially for Malacostracan predators, and a lesser influence of Class. Different colours in (a) are used only for illustrative purposes to indicate different species groups.

films (28%), spheres (15%) and other miscellaneous forms (foams and 'other', 5%) (Supplementary Table 2). The size of microplastics ranged from a 0.2 µm fragment (in the polychaete *Hediste diversicolor*, Tunisia) to a 17 cm fibre (in the decapod *Nephrops norvegicus*, Irish Sea). The most common colour was blue (28%), followed by clear (20%), black/grey (16%), white (14%), red (11%) and miscellaneous colours (greens, oranges, yellows and 'other', range 2–4%). The predominant polymers were cellulose (37%), polyethylene terephthalate (16%), polyethylene (12%), polyamide nylon (8%), polypropylene (7%) and 10 other polymers (range, 0.2–4%) (Supplementary Table 2). There were differences in the mean size (± s.e.) of microplastic particles between trophic groups ($F_{5,224} = 2.35$, $p = 0.042$; Fig. 5A), with herbivores retaining the largest microplastic particles (1362.6 ± 123.8 µm) followed by predators (1180.3 ± 160.1 µm), deposit feeders (840.5 ± 73.2 µm) and omnivores (799.3 ± 150.7 µm). There was no strong evidence that the

size of microplastic related to individual body size (Pearson's correlation: wet weight (g), r (226) = −0.05, $p = 0.409$, $n = 214$; body size (mm), r (212) = 0.29, $p < 0.001$, $n = 214$), indicating that species handle a range of microplastics (Fig. 5a). We also found differences in microplastic shape between trophic groups; predators and deposit feeders primarily retained fibres, whilst suspension feeders retain fibres and fragments (ratio of fibres:fragments; predators, 14.3:1; deposit feeders, 20.5:1; suspension feeders, 2.37:1; Fig. 5b).

The first principal component (PC1) explained 86.6% of the variation in the data and has a strong association with fibres and fragments (Eigenvectors: 0.759 and −0.639, respectively), with PC2 explaining 12% driven predominantly by sphere data. Fragments, whilst found in 66% of observations comprised, on average, 17% (± 1.16%) of the individual microplastic body burden. Very few individuals were burdened with spheres (≤4%, ± 0.71%) of microplastics ingested), films (3%, ± 0.38%), pellets (2%, ± 0.15%), foams (0.1%, ± 0.02%), or 'other' shapes (0.01%, ± 0.01%) (Supplementary Table 2).

## Discussion

Sea surface microplastic concentrations are known to vary by at least 6 orders of magnitude across the global ocean[46], with the frequency of species-microplastic interactions often suggested (e.g.[47,48]), or explicitly linked to (e.g.[47]) these environmental levels. Our analyses of the global inventory of marine invertebrate microplastic body burden reveal that whilst geographical location contributed most to the likelihood of uptake for an organism, it only explains 11% of the data variability, with Class and feeding mode each explaining a further 6% of the data. This highlights the importance of local, yet undetermined, environmental correlates and/or the role of organism-sediment interactions in setting and influencing microplastic bioavailability[49]. Filter feeders are often presented as the functional group most likely to be at risk of microplastic ingestion[50,51] due to the high volumes of water they process and the indiscriminate nature of filter feeding[52]. Such evidence, however, is based on a limited range of species and/or polymer types[23,53,54]. Further, there is a growing body of evidence to suggest that the sorting and rejection of particles of specific size ranges or polymers by filter-feeding bivalves may skew uptake data towards the preferred particle size for a given species[55,56], and experimental designs do not always consider the natural setting or the organisms being tested[57–59]. Our analyses applied to the full global dataset do not support the view that filter feeders are more prone to microplastic uptake than other species, likely due to mechanistic differences between species that alter microplastic capture and retention rates relative to other groups[60].

Compared to other groups, omnivores, predators and deposit feeders had much higher body burdens of microplastic. It is reasonable to consider therefore, that the bioaccumulation of microplastic is associated with predatory and omnivorous lifestyles[61]. Evidence for bioaccumulation of microplastic is not well supported empirically, with very little additional evidence for the translocation of microplastics into tissues; a pre-requisite for bioaccumulation[62]. Previous laboratory-based work has suggested that lower trophic level benthic organisms are at greater risk from microplastic exposure[30], but the higher body burdens in secondary consumers identified here suggest the contrary. Our findings give credence to the view that more subtle functional trait descriptors[63] are required to explain the mechanistic processes that determine uptake. Whilst the full risks of accumulating microplastics remain to be determined, elevated microplastic body burden remains an important measure of vulnerability[30,31,64]. Consequently, whilst changes in environmental microplastic contamination levels will have cascading effects on the microplastic body burden of some species, the relative uptake of microplastic will reflect the functional role and feeding modes of species within a community.

Recommendations for environmental monitoring of marine microplastics often suggest that particles should be measured within

defined size categories and be classified according to shape and polymer type as a key component of assessing the risk that they pose to wildlife[9,65,66]. The relevance of these particle characteristics for risk assessment has been well demonstrated within controlled laboratory ecotoxicology studies (e.g.[10,67–69]). Here, our analyses revealed that the bulk of microplastic particles present within benthic invertebrate species sampled globally were fibres. High numbers of fibres are consistent with the seawater and sediment microplastic data for coastal settings[15] but, acknowledging that positive[70] or negative[71] detection bias cannot be ruled out, these findings do lend further credence to the view that certain lifestyle characteristics can be important in determining uptake, particularly if morphology and shape of a microplastic particle alters the potential to be retained within an individual relative to other shapes through selective feeding or differences in particle handling. An alternative explanation may relate to gut morphology and the structural and functional complexity of the microplastic particle. Species with differentiated or much-folded digestive anatomy, such as crustacean amphipods and decapods[18,72–74], were strongly associated with an increased body burden of microplastic fibres, indicating that fibrous material becomes entangled and/or concentrated in morphologically complex anatomical features (Fig. 5b)[74–78]. The gastric mill in crustacea, in particular, can act to shred fibres causing entanglement, leading to higher levels of burden[73,79]. Retention of microplastic, however, also depends on the ability to excrete microplastics[80,81]. Gut transit times vary due to the presence and quality of food (e.g. in sea urchins)[82], the size of an organism (food takes longer to transit a longer gut)[83], as well as the complexity of an organism's gut morphology[84].

There is a weight of evidence from laboratory ecotoxicology studies that interactions with microplastics can impair survival, development, reproduction, growth and feeding[3,30,31,85] and alter biogeochemical processes in the sediment[86,87]. Hence, understanding which, and when, organisms may be at greater risk from microplastic exposure will help with modelling the ecosystem consequences of microplastic contamination into the future. The lack of correlation between individual microplastic body burden and environmental position (demersal, infaunal, and epifaunal) aligns with findings reported elsewhere (reviewed in Bour et al.[20]), and suggests that more subtle processes, such as, food availability[88,89], biofilm formation[90,91], and interactions with species (predator-prey)[49] or the sedimentary habitat, determine microplastic uptake[14,92]. Few studies (7% of those considered here) reported microplastic body burden alongside environmental contamination levels, so partitioning between organisms and their environment is not possible. Further, the paucity of studies that measure the excretion rates of organisms limits our understanding of net exposure (the number of particles passing through an organism over time) that organisms experience in their natural settings.

This analysis emphasises significant gaps in knowledge about the distribution of microplastic. Completion of a global inventory of microplastic contamination will require the development of more rapid, high throughput methods combined with models that incorporate environmental correlates and important aspects of species behaviour that alter the likelihood of microplastic uptake. Here, we propose a hierarchy for prioritising future research, identifying species groups most at risk from enhanced microplastic body burdens. First, a list of species should be identified for the geographical region of interest, after which taxonomic screening could be used to target those species most at risk, guided by the outcomes of this analysis. Combining these data with faunal behaviour, environmental context, and physiological factors, such as excretion rate, will hasten understanding of the mechanistic processes that determine net uptake in marine benthic invertebrates. Only by adopting a holistic view of ingestion, retention and/or excretion mechanisms that considers individual particle toxicity[9], species-specific sensitivity to microplastic

and associated contaminants[16], and the likelihood of exposure, will progress be made in determining the true extent of bioaccessibility and, ultimately, the risk of microplastics.

## Methods
We adopted the PICO method[93] to develop a search strategy for our analysis. Briefly, we used the Clarivate Web of Science (https://www.webofknowledge.com/) to source peer-reviewed articles that contained measurements of microplastic body burden using specific search terms (listed in Supplementary Note 1).

### Eligibility criteria
To be included in our quantitative synthesis, each study had to meet the following criteria: (1) an empirical study focussing on marine associated and benthic dwelling invertebrate species; (2) the focus was on microplastic exposure; (3) the organisms must be field collected organisms rather than laboratory studies; (4) the study must report particles per individual or particles per gram of wet weight of tissue; (5) evidence of quality assurances such as contamination control and spectroscopic confirmation of plastic presence were required. For more detail on the PICO exercise and our inclusion and exclusion criteria see Supplementary Table 1.

### Data curation and manipulation
Due to the wide range of reporting metrics and styles, several data manipulations were necessary for the data recovered from our search, and additional data were curated to complete the dataset, detailed as follows. The taxonomy for all species was aligned to current accepted status using the World Register of Marine Species 'Match Taxa' utility (https://www.marinespecies.org/). Sampling locations (latitude/longitude) were taken from the manuscripts and when not provided, the location of the samples was approximated using Google Maps[94] utilising the available information (descriptions and maps) in the manuscript. When small numbers of individuals of a species in close proximity were reported, or the reporting of data in the manuscript represented a total population, study sites were pooled and the data averaged.

As the biomass of an individual does not necessarily reflect an organism's functional biology[95,96] (Fig. S2), or correlate with microplastic body burden[18,19] (Fig. S3), we express microplastic burden as microplastic particles per individual (MP ind$^{-1}$) and we treat biomass as a trait. This avoids standardising microplastic burden by whole organism biomass, which assumes all biomass contributes to microplastic uptake or retention, and allows us to test for an overall effect of biomass in our analyses. Individual microplastic body burden was derived from the mean number of reported microplastics for each species record, regardless of the number of individuals reported in each observation (range, 1 – 481 individuals).

The primary data needed for our analyses was the body burden of microplastic particles. In the (micro)plastic literature, ingested particles are generally reported as MP ind$^{-1}$ or items gram tissue$^{-1}$ (wet weight). In cases where only items per gram of wet weight was recorded, the authors were contacted for these data and, if not available, the numbers per individual were calculated using the average items per gram and multiplying by the wet weight of the species. Some studies reported MP ind$^{-1}$ for only those organisms that had ingested plastic. As this is a poor representation of the data, these findings were adjusted to MP ind$^{-1}$ for the total population sampled. Ash-free dry weight or dry weight were converted to wet weight using published conversion factors[97], following consultation with authors, or were inferred from the wider literature using, where possible, data from the same geographic region and family level.

All microplastic sizes were converted to micrometers and minimum, maximum and mean values were collected where reported. Where ranges were reported the median value was recorded and

where percentages, size bins, or less than/more than values were reported these size data were excluded.

Microplastic colours and polymer types were all converted to the percentage contribution to the total amount of plastic recovered due to the variety of reporting methods. When reporting colours in groups, these were divided up to an equal representation. For instance, when reporting 'red, blue and green particles made up 30% of the total' each colour was scored as 10%.

Where data were not presented, or were not provided by authors, data were extracted from figures by A.P. using online software WebPlotDigitizer[98].

### Trait attribution
Trait categories most likely to influence the likelihood of microplastic ingestion by benthic invertebrate species were selected using the BIOTIC framework produced by The Marine Life Information Network for Britain and Ireland[99]. Specifically, we used the categories weight (size), environmental position, feeding mechanism, mobility, and habit.

### Estimating levels of environmental contamination
Contamination of the sediment was rarely (12/55 studies) measured at the same time as sampling the organisms. We used a global ocean surface microplastic contamination model[100] as a predictor of microplastic contamination of the benthic realm. The model is designed for sea surface contamination, but as all our records were taken in shelf sea regions, we assume here that surface contamination is reflective of the contamination levels of the seabed[101] as the sinking rate of particles in this model are consistent with observations of microplastic sinking rates[102–104]. The average level of contamination in particles $km^{-2}$ was calculated for all points falling within 250 km of a record using the van Sebille model[100] ($1° \times 1°$ resolution) and using ArcGIS Desktop[105] to help smooth the interpolated model, although we acknowledge that high levels of spatial heterogeneity can occur[106] due to horizontal separation of particles associated with biogeographical context[107,108], and particle size class[109], density[104], and morphology[102,110].

We ran linear models to investigate the relationships between the number of microplastics individual[-1] and the mean microplastic contamination predicted by the Van Sebille model within a 250 km radius of the sampled organism record, Longhurst Provinces, Spalding's Ecoregions (using realms and provinces as independent variables), and Class (Fig. S4). The van Sebille model, and Longhurst provinces were not capable of explaining our geographic trends in contamination levels but Spalding's provinces, realms and Class were. As provinces provided a greater number of geographic areas and still had significant explanatory power these were used alongside Class in our analyses. Other levels of taxonomy were explored but they either did not provide enough explanatory power or the number of data points was insufficient at these lower taxonomic levels to form a rigorous statistical analysis.

Using the Web of Science database, we identified 1519 studies, of which 412 were relevant to our research question. Using a strict set of inclusion and exclusion criteria (see Supplementary Table 1) we identified 55 studies that contained data in a usable format. The data span 16 provinces as defined by Spalding[36,37] (Fig. 1, generated using Shapefiles from Flanders Marine Institute[111]) with 90% of the records located in the Northern Hemisphere, predominantly across Europe and Southeast Asia.

### Data analyses
We focussed our analysis on Classes with a minimum of 30 observations: Bivalvia ($n = 192$), Malacostraca ($n = 69$), Gastropoda ($n = 54$) and Polychaeta ($n = 31$). MP ind[-1] were grouped based on Spalding's classification of biogeographical provinces[37] to allow regional assessment of MP ind[-1] [112]. To determine the relationship of MP ind[-1] Class[-1] between provinces we used a two-way ANOVA with Class (4 classes) and Province (15 provinces) as categorical explanatory variables. As the graphical model validation procedure (residuals vs fitted values and QQ-plot to assess the homogeneity of variances and normality) indicated that the statistical assumption of homogeneity of variance was not met, we continued with a generalised least-squares (GLS) estimation procedure that incorporates a variance-covariance term (using varIdent for categorical variables) to model the variance structure[113]. To determine the optimal structure in terms of random components we used restricted maximum-likelihood (REML) estimation and compared the model without a variance-covariate term to alternate models, including either Class or Province as variance-covariates using AIC and validation of model residual patterns. The optimal fixed-effects structure was then determined by backward selection using a likelihood ratio test obtained by maximum-likelihood (ML) estimation[114]. All analyses were performed using the nlme package[115] in the R statistical programming environment (v 4.1.2, R Core Team[116]).

To understand whether biological traits affect microplastic body burden, we used a linear model, and the entire dataset, with wet weight, environmental position, feeding mode, mobility, and habit as the explanatory variables and MP individual[-1] as the independent variable. Insignificant variables were removed by backward selection and and the comparison of AIC.

The relative contribution of our significant variables was determined by calculating the proportion of data explained by each variable (scaling the sum of squares by 1 and dividing by their sum). Tukeys HSD tests were performed using the agricolae package[117] in R.

We used Pearson's correlation (r) to assess whether organism size (mm) or wet weight (g) influenced the size or number of microplastics observation[-1]. Principle component analyses were carried out in PRIMER (version 6.3.13)[118].

### Reporting summary
Further information on research design is available in the Nature Portfolio Reporting Summary linked to this article.

## Data availability
The research data supporting this publication are openly available from Harvard Dataverse at: https://doi.org/10.7910/DVN/E57LOA. The data for the Van Sebille 2015 model can be found at: https://figshare.com/collections/data_of_Van_Sebille_et_al_2015_ERL_paper/5764184. Ocean boundaries (Spalding's Provinces) used in Fig. 1 are freely available at: https://www.worldwildlife.org/publications/marine-ecoregions-of-the-world-a-bioregionalization-of-coastal-and-shelf-areas. Longhurst provinces used as a geographical variable in the initial analysis are freely available from: https://www.marineregions.org/gazetteer.php?p=details&id=22538. The world country shapefiles used in Fig. 1 are available from ESRI at: https://hub.arcgis.com/datasets/esri::world-countries-generalized/about and available for use under the ESRI Master License Agreement. Taxonomy for all species was verified and curated using the World Register of Marine Species match taxa function available at: https://www.marinespecies.org/aphia.php?p=match. Biological trait categories were modified using those provided by the Marine Life Information Network (MarLIN) Biological Traits Information Catalogue (BIOTIC) available at: https://www.marlin.ac.uk/biotic/resources.php. Latitudes and Longitudes when not specifically mentioned in the individual study were approximated using Google Maps.

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

## Acknowledgements

The authors would like to thank our colleagues at the Centre for Environment, Fisheries and Aquaculture Science (CEFAS) for their counsel in the initial conception of this work and to M. Curtis, P. McIlwaine and S. Ware in particular for their assistance with collating species weights missing from publications. We also want to thank the Galloway and Lewis lab group: Lara, Clara, Alice, Kat, Paul, Jake, Jen, Steve, Steph, Daisy, Francisca, and Emily for their support. The authors also want to thank all those who supplied additional information from their own research needed to complete the original dataset and in particular T. Souster (BAS), T. Palmer (Texas A&M), and Å.I. Wilhelmsen (NHM Oslo) for filling in data gaps with their expert knowledge. AP, TG, CL, JAG, and MS would like to acknowledge support from NERC grant NE/S003975/1.

## Author contributions

A.P.: Conceptualization, Methodology, Validation, Formal analysis, Investigation, Data curation, Writing – original draft, Writing – review & editing, Visualization. J.A.G.: Methodology, Validation, Formal analysis, Writing – review & editing, Visualization, Funding acquisition. G.S.: Methodology, Validation, Data curation, Writing – review & editing. C.N.L.: Conceptualization, Validation, Writing – Review & Editing, Supervision, Funding acquisition. M.S.: Methodology, Validation, Data curation, Writing – review & editing, Funding acquisition. T.G.:

Resources, Writing – review & editing, Supervision, Project administration, Funding acquisition.

## Competing interests
The authors declare no competing interests.
