## [Peer Review File · Nature Communications]

Microplastic burden in marine benthic invertebrates depends on species traits and feeding ecology within biogeographical provincesREVIEWER COMMENTS

Reviewer #1 (Remarks to the Author):

My thanks to the authors and editor for the chance to view this interesting and enjoyable paper, the results of which add to our understanding of the factors influencing the microplastic body-burden of invertebrate taxa. The methods (both the review and analytical steps) are well outlined; however, I do have a query regarding the suitability of sea surface microplastics averaged over an apparently large radius ("The average level of contamination in particles per km² was calculated for all points falling within 250km of a record using the van Seville model") to seabed microplastics. Although the issue of the disconnect between water column plastics is addressed, I believe there is insufficient consideration given in the text to the high heterogeneity of microplastic contamination in benthic habitats and potential sources of variation in plastic settlement and representation. Nootboom's modeled diatoms sank at a rate of "6 m/day between 10- and 100-m depths and... between 100 and 2,000 m up to 45 m/day" or as previous but increasing "2,000- and 3,500-m depths up to 65 m/day" with lower transport observed in relating to faster sinking and aggregate formation. How does this relate to both modelled and observed sinking rates of microplastics? How might the shape and density of these polymers alter this and the resulting distribution? We know that microplastic concentrations at depth demonstrate high heterogeneity, but how might we expect the potential sed. microplastic concentration to vary over the same geographic scales from which you have obtained your surface level averages? I'd also have liked to see a little bit more about the background data behind your averaged points, as well as the linear analysis results excluding different geographic factors, in your supplementary material.

In addition, I have a number of comments in the text, including:

Figure 1: Great figure, but you can't see the detail. Perhaps a panel showing the major ocean basins. Full sized map at the top, then either zoomed to basin scale below or an Atlantic slice and an indo-pacific slice. Get rid of some of that pesky terrestrial whitespace.

Figure 3: can you remove the grey background on the chart and make the icons in your class key larger (I know ggplot is occasionally a pain!). Additionally, please check your graphs for clarity if printed in greyscale (there's nothing worse than trekking to the color printer), and while we're here, the point size of your fonts could be larger on this and your other images (I currently have my PDF at 158% to get a decent view).

Figure 4: this graph is mostly whitespace.

Figure 5A: again, mostly white space

Line 26-29: Two "but"s in the same sentence, consider revising.

Line 29: "examine" seems to imply a physical element. Meta analysis? Integrated existing data? Explore?

Line 40: You might need to be clearer about how this information helps "prioritise waste management strategies". To what extent can this information help to identify sources and deliver actionable goals beyond, as an example, "what are we going to do about all of these fibres?"

Line 73: Missing a comma after "thresholds" and "solutions"

Line 76-81: Can you break this down into smaller sentences (or pop "with taxonomic information" into a set of commas) here?

Line 83: A pedant's question, but "uncertainty" regarding? Are you linking back to 71-76 here?

Line 94: Not sure about the phrasing of this line, you're only revealing the gaps, the original authors are the ones that revealed their records. Perhaps a "Despite records... across 16

provinces, our analysis revealed..."

Line 117: Is this "risk of uptake", or is it degree of retention?

Line 152: Was there scope to do any within class analysis? I can see you have broken down by order and family in the spreadsheet.

Line 195 – 197: May this be an artifact of reduced retention in this species?

Line 321: "also depends on the ability to excrete". The comments before this also link to ability to excrete plastics. Perhaps this statement should appear earlier?

Line 463: Highlight your reasoning in including Longhurst Provinces and Spalding's Ecoregions (using realms and provinces as independent variables) as "Controlling for Environmental Contamination"?

Reviewer #2 (Remarks to the Author):

The manuscript titled "Assessing Microplastic Body Burden in Marine Invertebrates: The Role of Species-Microplastic Interactions" explores the relationship between microplastic contamination in marine sediments and the microplastic body burden of benthic invertebrate species across various biogeographic areas. There is a widely held assumption that microplastic body burdens in marine organisms reflect the levels of environmental contamination. This often ignores the important role of species-specific traits, feeding habits, and behavior in microplastic ingestion. Through their meta-analysis, the authors show that the absolute levels of microplastic contamination in the abiotic compartment is not the only determinant of body burdens. The authors point out that the highest microplastic body burdens are found among omnivores, predators, and deposit feeders, rather than suspension feeding species which challenges prevailing wisdom. These are noteworthy findings and will be of use to microplastics research community.

The manuscript is well-written and methodology is appropriate for the meta-analysis.

Consequently, the outcomes are nuanced, but not taken too far. There are some specific instances where I think the authors need to clarify their thinking or provide additional information.

Specific comments

Use of the term "uptake". Throughout the manuscript the authors refer to the "uptake" of microplastics. This to me is highly debatable terminology. In solute toxicology, "uptake" would refer to the process of a contaminant crossing a biological membrane at the gill or gut (i.e. a site of uptake). For MPs, the term is often used simply (and wrongly, in my opinion) to infer ingestion or retention of the MP within the digestive tract and not the actual uptake of MPs from the gut. The authors do define species-microplastic interactions as "adhesion, entanglement, and ingestion" using ingestion rather than uptake (L72) and then as a factor of "gut retention times" (L83). For me these are more accurate terms than uptake which is not demonstrated in many of the papers in the dataset. The authors should either qualify their use of "uptake" early in the manuscript if they do indeed mean uptake, or use terms like microplastics ingestion or retention.

Specific instances include the following, but I urge the authors to check their terminology around uptake carefully.

L. 113. "Microplastic uptake was common" should be "The presence of microplastics was common"

L353. "uptake and depuration" could more accurately read as "ingestion, retention and/or excretion" as it is not entry and loss from the tissue but rather entry and expulsion from digestive processes that is in focus.

In the discussion about fibres (L308). It would be worth discussing that a number of recent studies focussing on fibres have shown that not all fibres are of synthetic origin and natural fibres (wool, cotton) are also a significant proportion of what is included as microplastic fibres:

KeChi-Okafor, C., Khan, F. R., Al-Naimi, U., Béguerie, V., Bowen, L., Gallidabino, M. D., ... & Sheridan, K. J. (2023). Prevalence and characterisation of microfibrils along the Kenyan and Tanzanian coast. *Frontiers in Ecology and Evolution*, 11, 1020919.

Gallidabino, M., Sheridan, K., Stanton, T., James, A., & Ginting, J. J. (2023). Are microfibrils a

problem for aquatic ecosystems? What we don't know about textile pollution. Synthetic fibres, 72, 64.

L. 343-346. A global inventory requires high throughput method, but also a harmonized approach to produce comparable dataset across the world.

The resolution of Figure 1 seems rather low. Please improve.

Response to reviewers' comments for Porter et al.; *Species traits and feeding ecology within biogeographical provinces moderate microplastic burden in the marine benthos* (NCOMMS-23-38530-T).

Reviewer #1:

1. I do have a query regarding the suitability of sea surface microplastics:
 - a. averaged over an apparently large radius ("The average level of contamination in particles per km² was calculated for **all points falling within 250km of a record using the van Seville model**") to seabed microplastics. Although the issue of the disconnect between water column plastics is addressed, I believe there is insufficient consideration given in the text to the high heterogeneity of microplastic contamination in benthic habitats and potential sources of variation in plastic settlement and representation.

We agree with the reviewer that vertical transport is key to horizontal transport because variations in the 3D velocity field will determine where each size class reaches the seafloor. Such information is, however, very poorly constrained or unknown. As the reviewer has noted, we have addressed this surface-seafloor disconnect by using the van Seville model, which standardises the spatial distribution of plastic marine debris at 1 x 1 km using three ocean circulation models. This is the best currently available model. In contrast to previous work, which has aggregated observations at much larger scales/lower resolution (e.g. mean particles per Longhurst region, Savoca et al. 2021, DOI: 10.1111/gcb.15533), we follow best practice (following Compa et al. 2019, DOI: 10.1016/j.scitotenv.2019.04.355) and use the mean particle density within a 250km radius to stabilise variability.

Nevertheless, we agree with the reviewer that it would be remiss of us not to highlight the spatial variability that exists in the global inventory of plastic burden observations (our dataset is the most comprehensive collation to date). We have now acknowledged the issue of spatial heterogeneity in the Methods section and included appropriate supporting references on lines 465 – 468.

- b. Nootboom's modeled diatoms sank at a rate of "6 m/day between 10- and 100-m depths and... ..between 100 and 2,000 m up to 45 m/day" or as previous but increasing "2,000- and 3,500-m depths up to 65 m/day" with lower transport observed in relating to faster sinking and aggregate formation. How does this relate to both modelled and observed sinking rates of microplastics? How might the shape and density of these polymers alter this and the resulting distribution? We know that microplastic concentrations at depth demonstrate high heterogeneity, but how might we expect the potential sed. microplastic concentration to vary over the same geographic scales from which you have obtained your surface level averages?

The reviewer is highlighting that the model we have used is based on the sinking rate of diatoms rather than plastic. Our own previous work on the sinking rates of microplastics (Porter et al. 2018, DOI: 10.1021/acs.est.8b01000) has shown that microplastics can sink at rates between 0.3 - 354m day⁻¹, but also that plastic particles do not necessarily behave independently as they aggregate into marine snows (sinking at ~500-900 m day⁻¹). We are satisfied that these rates are consistent with those within Nootboom's model. It is also important to emphasise that the majority (97%) of plastic burden observations within our study are from shallower waters (shelf depth, ≤ 200m), where

sinking rate is less critical and horizontal spread tends to be more constrained than at the greater water depths of the open ocean.

The reviewer is correct that the different polymers and shapes of plastic will affect their sinking rate in different ways, and that plastics in the marine environment may aggregate with one another.

We now include these important points about sinking rate and particle morphology in the methods section, supported by appropriate supporting references on lines 466 – 468 and thank the reviewer for pointing out this omission as it further endorses our approach.

- c. I'd also have liked to see a little bit more about the background data behind your averaged points, as well as the linear analysis results excluding different geographic factors, in your supplementary material.

The information the reviewer requests, and our full dataset, is provided in an open source repository (Harvard Dataverse) linked to this contribution. We have added the location of the dataset and its DOI to the acknowledgements. We are happy to provide a supplementary figure of the linear analysis results that excluded the geographic factors the reviewer is referring as a new supplementary figure in the Methods section (Figure S4).

2. Figure 1: Great figure, but you can't see the detail. Perhaps a panel showing the major ocean basins. Full sized map at the top, then either zoomed to basin scale below or an Atlantic slice and an indo-pacific slice. Get rid of some of that pesky terrestrial whitespace.

We thank the reviewer for this suggestion. We have modified Figure 1 to include two additional basin scale panels for the NE Atlantic & Mediterranean, and SE Asia as they are the regions of most interest. We prefer to retain the landmass as whitespace, as exclusion would distort the geographical layout of the figure.

3. Figure 3: can you remove the grey background on the chart and make the icons in your class key larger (I know ggplot is occasionally a pain!). Additionally, please check your graphs for clarity if printed in greyscale (there's nothing worse than trekking to the color printer), and while we're here, the point size of your fonts could be larger on this and your other images (I currently have my PDF at 158% to get a decent view).

We have adjusted Figure 3 in line with the reviewers suggestions.

4. Figure 4: this graph is mostly whitespace.

This figure is a standard format mean \pm s.e. plot. The white space will be less prominent when in final typeset form.

5. Figure 5A: again, mostly white space

See response to point 4.

6. Line 26-29: Two "but"s in the same sentence, consider revising.

Corrected.

7. Line 29: “examine” seems to imply a physical element. Meta analysis? Integrated existing data? Explore?

Corrected.

8. Line 40: You might need to be clearer about how this information helps “prioritise waste management strategies”. To what extent can this information help to identify sources and deliver actionable goals beyond, as an example, “what are we going to do about all of these fibres?”

We make a general point here in the abstract, but already devote a paragraph to this point in the final paragraph of the main manuscript.

9. Line 73: Missing a commas after “thresholds” and “solutions”

We have made this correction.

10. Line 76-81: Can you break this down into smaller sentences (or pop “with taxonomic information” into a set of commas) here?

Corrected.

11. Line 83: A pedant’s question, but “uncertainty” regarding? Are you linking back to 71-76 here?

We have revised the sentence to clarify our meaning.

12. Line 94: Not sure about the phrasing of this line, you’re only revealing the gaps, the original authors are the ones that revealed their records. Perhaps a “Despite records... ..across 16 provinces, our analysis revealed...”

The reviewer seems to have misunderstood the statement; the two halves of the sentence follow on from each other but are separate statements. We are reporting what our analysis revealed, rather than pointing out the limitations of the global dataset. No change is necessary.

13. Line 117: Is this “risk of uptake”, or is it degree of retention?

The use of 'uptake' is commonplace in microplastic science to encompass the combined contributions that both external adhesion (i.e. on gills and external feeding structures) and ingestion make to the measured individual body burden at the time of sampling, since the majority of sampling and analysis methodologies cannot distinguish between these two processes. As such we prefer to use our original phrasing as the specific route of uptake into marine invertebrates is often not reported or is out of scope of the study methodologies within the source material of this meta-analysis. The term 'uptake' is widely used within the field for precisely this reason, as highlighted by Reviewer 2, for example : Redondo-Hasselerharm et al., 2020 (Sci. Adv. 10.1126/sciadv.aay4054),

Katija et al., 2017 (Sci. Adv. 10.1126/sciadv.1700715), Law and Thompson, 2014 (Science 10.1126/science.1254065), Roch, Friedrich & Brinker, 2020 (Sci. Reps. 10.1038/s41598-020-60630-1). The term 'retention' implies that we understand excretion rates, which cannot be inferred from the available data and is poorly understood for the majority of species. We have therefore clarified our use of the term 'uptake' on lines 59-60 as with point 18 from Reviewer 2.

14. Line 152: Was there scope to do any within class analysis? I can see you have broken down by order and family in the spreadsheet.

We had hoped to perform analyses at levels lower than Class, however, the number of data points was insufficient at these lower taxonomic levels to form a rigorous statistical analysis. Class was the lowest level of taxonomy we could analyse. We have added this in the methods to explain more fully.

15. Line 195 – 197: May this be an artifact of reduced retention in this species?

No. The reduction in particles per individual following the removal of the outliers relates to the calculation of the mean without these influential points. We include means with and without these outliers to illustrate the disproportionate role of these species.

16. Line 321: "also depends on the ability to excrete". The comments before this also link to ability to excrete plastics. Perhaps this statement should appear earlier?

We appreciate this suggestion and have altered the text earlier (on line 323) to remove the reference to retention. This we believe makes the point unfold in the correct sequence and therefore the text is now appropriate where it is currently positioned.

17. Line 463: Highlight your reasoning in including Longhurst Provinces and Spalding's Ecoregions (using realms and provinces as independent variables) as "Controlling for Environmental Contamination"?

The meaning was incorrect here. We have adjusted the section title to "Estimating levels of environmental contamination" to avoid misinterpretation (Line 455).

Reviewer #2:

18. Use of the term "uptake". Throughout the manuscript the authors refer to the "uptake" of microplastics. This to me is highly debatable terminology. In solute toxicology, "uptake" would refer to the process of a contaminant crossing a biological membrane at the gill or gut (i.e. a site of uptake). For MPs, the term is often used simply (and wrongly, in my opinion) to infer ingestion or retention of the MP within the digestive tract and not the actual uptake of MPs from the gut. The authors do define species-microplastic interactions as "adhesion, entanglement, and ingestion" using ingestion rather than uptake (L72) and then as a factor of "gut retention times" (L83). For me these are more accurate terms than uptake which is not demonstrated in many of the papers in the dataset. The authors should either qualify

their use of “uptake” early in the manuscript if they do indeed mean uptake, or use terms like microplastics ingestion or retention.

We agree with the reviewer and have followed their recommendation to qualify our use of uptake early in the manuscript (Line 59 – 60). We now define what we mean at the first usage of the term in the main body of the text. The suggestion to use the alternate terms of ‘ingestion’ or ‘retention’ are equally difficult, as ‘ingestion’ ignores external adhesion and ‘retention’ implies a knowledge of ingestion/egestion rates.

19. L. 113. “Microplastic uptake was common” should be “The presence of microplastics was common”

Corrected.

20. L353. “uptake and depuration” could more accurately read as “ingestion, retention and/or excretion” as it is not entry and loss from the tissue but rather entry and expulsion from digestive processes that is in focus.

Corrected in line with reviewers suggestion.

21. In the discussion about fibres (L308). It would be worth discussing that a number of recent studies focussing on fibres have shown that not all fibres are of synthetic origin and natural fibres (wool, cotton) are also a significant proportion of what is included as microplastic fibres:

KeChi-Okafor, C., Khan, F. R., Al-Naimi, U., Béguerie, V., Bowen, L., Gallidabino, M. D., ... & Sheridan, K. J. (2023). Prevalence and characterisation of microfibrils along the Kenyan and Tanzanian coast. *Frontiers in Ecology and Evolution*, 11, 1020919.

Gallidabino, M., Sheridan, K., Stanton, T., James, A., & Ginting, J. J. (2023). Are microfibrils a problem for aquatic ecosystems? What we don't know about textile pollution. *Synthetic fibres*, 72, 64.

We thank the reviewer for this suggestion. We now include this point and support with the recommended KeChi-Okafor et al. citation.

22. L. 343-346. A global inventory requires high throughput method, but also a harmonized approach to produce comparable dataset across the world.

We sympathise with the reviewer's opinion on harmonisation, but an agreed international standard methodology has not yet been formally agreed. This is a live and active debate that is ongoing within the microplastics community (e.g. Mitrano et al., 2023. DOI:10.1021/acssuschemeng.3c03221; Pimpke et al., 2020. DOI: 10.1177/0003702820921465) and, although relevant, forms a distraction from the focus of our manuscript.

23. The resolution of Figure 1 seems rather low. Please improve.

We have modified Figure 1 to include two additional basin scale panels for the NE Atlantic & Mediterranean, and SE Asia as they are the regions of most interest.

Ends.

REVIEWERS' COMMENTS

Reviewer #1 (Remarks to the Author):

I'd like to take this opportunity to repeat my previous comments regarding the high level of interest and "readability" of this paper and its results, and can confirm that am happy that my comments have been fully addressed.

Reviewer #2 (Remarks to the Author):

The authors have answered the points raised by the Reviewers. I have not further comments and suggest that the manuscript can be accepted.

REVIEWERS' COMMENTS ON SECOND REVIEW

Reviewer #1 (Remarks to the Author):

I'd like to take this opportunity to repeat my previous comments regarding the high level of interest and "readability" of this paper and its results, and can confirm that am happy that my comments have been fully addressed.

Reviewer #2 (Remarks to the Author):

The authors have answered the points raised by the Reviewers. I have not further comments and suggest that the manuscript can be accepted.

Ends.